# Generalist Policy for $k$-Server Problem on Graphs using Deep Reinforcement Learning with Action-Value Decomposition

## Abstract

The *online $k$-server problem on graphs* is a fundamental computational problem that can model a wide range of practical problems, such as dispatching ambulances to serve accidents or dispatching taxis to serve ride requests. While most prior work on the $k$-server problem focused on online algorithms, reinforcement learning promises policies that require low computational effort during execution, which is critical in time-sensitive applications, such as ambulance dispatch. However, there exists no scalable reinforcement-learning approach for the $k$-server problem. To address this gap, we introduce a scalable computational approach for learning *generalist policies* for the $k$-server problem. Besides scalability, the advantage of generalist policies is transferability: a generalist policy can be applied to an entire class of graphs without the need for retraining, which is crucial for practical applications, e.g., in ambulance dispatch problems where road conditions or demand distributions may change over time. We achieve scalability and transferability by introducing a novel architecture that decomposes the action-value into a global and a local term, estimated from a shared graph-convolution backbone. We evaluate our approach on a variety of graph classes, comparing to well-established baselines, demonstrating the performance and transferability of our generalist policies.

## 1 Introduction

The $k$-server problem, originally proposed by Manasse et al. (1988; 1990), is one of the most fundamental problems from the perspective of mixed-integer programming, online algorithms, and competitive analysis (Bertsimas et al., 2019b). Koutsoupias (2009) claims that it is "perhaps the most influential online problem" in computer science. The problem setting is surprisingly simple—given a metric space or a weighted graph and $k$ mobile servers, the problem deals with moving the servers to the locations of *requests* that appear at arbitrary points in the metric space (or the graph) over time. At each time step, a request appears, and the decision-maker must immediately dispatch a server to serve the request. Usually, it is assumed that a new request arrives only after the previous request is served (Bertsimas et al., 2019b). The decision-maker seeks to minimize the distance traveled by the servers over time. There are two canonical versions of the problem—the *online* version, where the decision-maker must dispatch a server based on the current request and the past requests, and the *offline* version, where the decision-maker knows the set of requests (to be responded to) *a priori*. Despite the apparent simplicity of the problem setting, the $k$-server problem serves as a rich framework for in-depth analysis of many online problems (e.g., paging and caching (Bertsimas et al., 2019b), network analysis (Alon et al., 1995), dynamic resource allocation (Mukhopadhyay et al., 2016), and manufacturing (Privault & Finke, 2000)).

If the sequence of requests is known in advance, i.e., in the offline setting, the problem can be reduced to the standard network flow optimization problem and solved efficiently (Chrobak et al., 1991; Bertsimas et al., 2019b). The offline solution can then be used to gauge the efficacy of online approaches; indeed, a large part of prior work on solving the $k$-server problem focuses on establishing *competitive ratios* for the solution algorithms, which is defined as the worst-case ratio between the performance of an online algorithm and the optimal offline algorithm. One of the most well-known online approaches is the Work-Function Algorithm (WFA), which combines two

perspectives—a *strategic* view by considering past requests and a *tactical* greedy view by considering the current request. Many approaches have relied and built upon the central idea of WFA, e.g., the holistic adaptive robust optimization algorithm, also known as HARO (Bertsimas et al., 2019b) combines adaptive robust optimization techniques with WFA. It is worth pointing out that several simple algorithms that are not competitive (e.g., greedy algorithm that always dispatches the nearest server to a request) often work reasonably well in practice (Bertsimas et al., 2019b).

We focus on the stochastic $k$-server problem (Dehghani et al., 2017) where the arrival of requests at the nodes of a graph is governed by a *known* probability distribution.[1] Such a setting is largely motivated by practical problems of dynamic resource allocation under uncertainty, e.g., dispatching emergency responders to service distress calls and optimizing micro-transit fleets (Dehghani et al., 2017; Mukhopadhyay et al., 2016). The availability of the arrival distribution is fairly common in real-world applications. Consider the problem of micro-transit, where the decision-maker must optimize the allocation of taxis to passengers; in such scenarios, the arrival distribution can be empirically approximated by using historical data of passenger requests. Naturally, in real-world use cases, decisions must be made in an online manner, i.e., the exact sequence of requests is not known in advance. As a result, we focus on the online version of the problem.

We view the $k$-server problem as a sequential decision-making problem under uncertainty, where the decision-maker must optimize dispatch decisions over time to minimize the expected distance traveled by the servers; the expectation is taken with respect to a known probability distribution that governs the arrival of requests. We model the $k$-server problem as a Markov decision process (MDP). Our goal is to leverage the known arrival distribution and the graph topology to compute an optimal policy (i.e., a mapping from an arbitrary *state* of the problem to a dispatch action) in an offline manner. Once the policy is learned, it can be used to make decisions online as requests arrive in a stochastic manner.

Crucially, instead of learning a policy on a given graph, we seek to *learn a generalist policy* for any graph topology (drawn from the same distribution as the training topologies) and any request arrival distribution (drawn from the same distribution as the training distributions). Learning a generalist policy is critical for practical applications; e.g., consider the problem of dispatching emergency responders on a road network. In such a situation, traffic conditions can alter the edges of the graphs; indeed, some roads might close down, altering the structure of the graph. While re-training a policy from scratch is computationally expensive, a generalist policy can be invoked in constant time during decision-making without additional training.

We point out that the $k$-server problem has been modeled as an MDP before; e.g., Lins et al. (2019a) transform the $k$-server model to a visual task problem and model it as an MDP, and Even-Dar et al. (2009) discuss the $k$-server problem in the context of online MDPs (although they relax the Markovian assumption). However, prior work has significant limitations, both in terms of scalability and generalizability. For example, the training approach used by Lins et al. (2019a) does not scale to graphs of non-trivial size, and the trained policies cannot be transferred to other problem instances, thereby requiring re-training if the graph topology changes. To the best of our knowledge, we make two seminal contributions: **1)** we present the first effort to directly model (i.e., without any transformations) the $k$-server problem as an MDP by simply encoding the available information at any stage of the problem as the state of the MDP; and **2)** we present the first generalist learning approach for the $k$-server problem, which can be applied to arbitrary topologies and request arrival distributions (drawn from the same distribution as the training instances).

Naturally, this modeling paradigm presents several challenges, a major bottleneck being an enormously large state space that requires extremely specific actions to avoid long-term degradation of performance. Additionally, as we show empirically, online approaches do not scale to large problem instances. We address this challenge by presenting **the first scalable reinforcement learning-based solution to the $k$-server problem**. Second, ensuring scalability for a generalist policy is a major bottleneck—during execution, the size of the graph could be significantly larger than training. We ensure **scalability to arbitrarily large problem instances by extracting local structural information embedded in the graph topology**, i.e., we hypothesize that local information is often a determinant of global actions in this problem, thereby making our approach scalable by construc-

---

[1]Our problem is slightly different from (Dehghani et al., 2017); however, the broader framework of stochastic $k$-server problems subsumes our problem setting.

tion. We reiterate that *scalability* (to large problem instances) and *transferability* (to any request arrival distribution or graph topology) are critical for practical applications. **To the best of our knowledge, despite (almost) four decades of prior work on the $k$-server problem, our work is the first to address these challenges**.

A key novelty of our approach is the **graph-theoretic, global-local decomposition of the action-value with a shared backbone**, which could be applied to other sequential decision-making problems on large graphs. Note that this is fundamentally different from dueling Q-networks, which decompose the action-value into state-value and action advantage, requiring regularization and complete state for advantage estimation Wang et al. (2016). The key benefit of our approach is the ability to train efficiently on small graphs, and then apply trained policies to arbitrarily large graphs.

The rest of this paper is organized as follows. Section 2 introduces the $k$-server problem on graphs, formulating it as an MDP, and defines optimal policies. Section 3 describes our proposed computational approach based on global-local decomposition of action values. Section 4 evaluates our proposed approach numerically, comparing to various baseline algorithms and policies on a range of graphs. Section 5 provides a brief overview of related work. Finally, Section 6 concludes the paper.

We provide the complete source code for all of our experiments as supplementary material; we will make the source code publicly available upon acceptance.

## 2 $k$-SERVER PROBLEM ON GRAPHS

### 2.1 PROBLEM SETTING

We begin by formally introducing the stochastic $k$-server problem on graphs (a summary of notation is presented as a table in the appendix). We consider a connected graph $\mathcal{G} = \langle \mathcal{N}, \mathcal{E} \rangle$, where $\mathcal{N}$ is the set of nodes and $\mathcal{E}$ is the set of edges. We let $n = |\mathcal{N}|$. We use $d(u, v)$ to denote the shortest path distance (i.e., number of edges) between two graph nodes $u$ and $v$, where $u, v \in \mathcal{N}$. The problem considers a total of $T$ discrete time steps, and we use $t$ to denote an arbitrary time step. Requests arrive on the nodes of the graph according to a known probability distribution $\boldsymbol{p}$, where $p_v$ denotes the probability of a request arriving at node $v \in \mathcal{N}$. We denote the sequence of requests by $\{\sigma^1, \sigma^2, \ldots, \sigma^T\}$, where $\sigma^t \in \mathcal{N}$ is the location (i.e., a node of the graph) of the request at time $t$. We use $\boldsymbol{x}^t = \{x_1^t, \ldots, x_i^t, \ldots, x_k^t\}$ to denote the locations of the $k$ servers in time step $t$, where $x_i^t \in \mathcal{N}$ is the location of the $i$-th server. For a list of symbols, please see Table 2 in the appendix.

### 2.2 MODEL

We model the $k$-server problem as a Markov decision process (MDP). An MDP can be described by the tuple $\langle \mathcal{S}, \mathcal{A}, \rho, r \rangle$ where $\mathcal{S}$ is the set of states, $\mathcal{A}$ is the set of actions, $\rho$ is the state transition function with $\rho_a(\boldsymbol{s}, \boldsymbol{s}')$ being the probability with which the process transitions from state $\boldsymbol{s}$ to state $\boldsymbol{s}'$ when action $a$ is taken, and $r$ represents the reward function with $r(\boldsymbol{s}, a)$ being the reward for taking action $a$ in state $\boldsymbol{s}$. In our setting, the MDP is defined as follows.

**States** The state $\boldsymbol{s}^t \in \mathcal{S}$ of the process at time step $t$ is the tuple $\langle \boldsymbol{x}^t, \sigma^t \rangle$, where $\boldsymbol{x}^t$ are the locations of the $k$ servers (before dispatching one of them to serve the request), and $\sigma^t$ is the location of the request. For the initial state $\boldsymbol{s}^0$, the tuple $\langle \boldsymbol{x}^0, \sigma^0 \rangle$ captures the initial locations of the $k$ servers and of the first request, both chosen uniformly at random. Server locations are chosen ensuring that no two servers occupy the same location.

**Actions** At each time step $t$, the agent can choose an action $a^t \in \mathcal{A}$ that involves moving one of the servers to serve the current request $\sigma^t$. The action space $\mathcal{A}$ is represented as $a^t \in \{1, 2, \ldots, k\}$ where $a^t = j$ specifies the action of moving the $j$-th server at time step $t$. In other words, the action $a_t$ denotes the index of the server that is moved to serve the request at time $t$. If the request arrives at the same location as one of the servers, the action is simply dispatching that server to fulfill the request.

**Reward Function** We model the reward function $r(\boldsymbol{s}^t, a^t)$ as the negative of the distance traveled by the chosen server to serve the request, i.e., $r(\boldsymbol{s}^t, a^t) = -d(x_{a^t}^t, \sigma^t)$, where $d(x_a^t, \sigma^t)$ is the distance between the location $x_{a^t}^t$ of the chosen server $a^t$ at time $t$ and the location $\sigma^t$ of the request. The distance between two nodes is measured as the number of hops between the nodes.

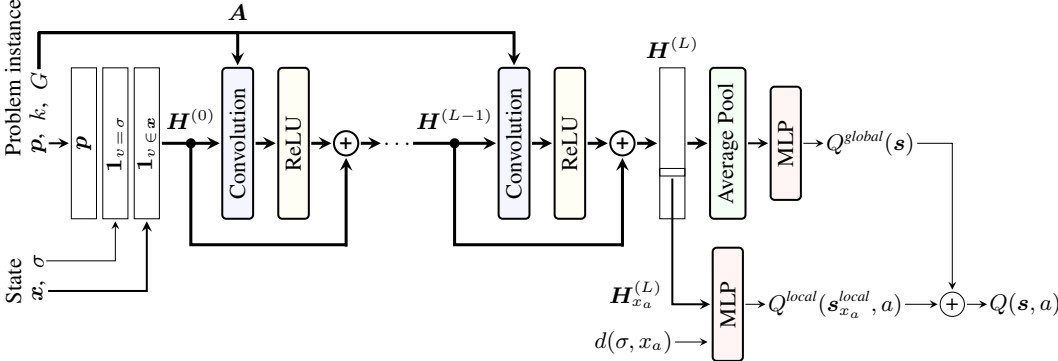

Figure 1: Overview of the architecture of our action-value estimator. Note the decomposition of action-value $Q(\boldsymbol{s}, a)$ into $Q^{global}$, which depends on the entire state but not the action, and into $Q^{local}$, which depends on both the state and the action but only on a local neighborhood around node $x_a$. Thick lines represent matrices; semi-thick lines represent vectors; thin lines represent scalar values.

**Transition Function** Given a state $\boldsymbol{s}^t = \langle \boldsymbol{x}^t, \sigma^t \rangle$ and an action $a^t$, two changes occur as the process transitions to time $t + 1$. First, the location of server $a^t$ changes to the location of the request that it is dispatched to serve, i.e., its location changes to $\sigma^t$. Second, a new request $\sigma^{t+1}$ arrives at random according to the probabilities $\boldsymbol{p}$, resulting in a new state $s^{t+1} = (\boldsymbol{x}^{t+1}, \sigma^{t+1})$. Note that $\boldsymbol{x}^{t+1}$ is identical to $\boldsymbol{x}^t$ except for its $a^t$-th element, which is $\sigma^t$.

**Optimal Policy** The decision-maker's goal is to find an optimal policy $\pi^*$. A policy $\pi : \mathcal{S} \to \mathcal{A}$ is a mapping of states to actions. An optimal policy maximizes the expected discounted rewards, i.e.,

$$\pi^*(\boldsymbol{s}^t) = \arg\max_{a^t \in \mathcal{A}} \mathbb{E}_{\boldsymbol{s}^{t+1}, a^{t+1}, \boldsymbol{s}^{t+2}, \dots} \left[ \sum_{\tau = t}^{\infty} \gamma^{\tau - t} \cdot r(\boldsymbol{s}^{\tau}, a^{\tau}) \right]$$

where future states and actions are drawn according to the state transition function $\rho$ and policy $\pi^*$. In our case, since we model rewards as the distance traveled by the servers, the agent actually accrues a *cost*, which we seek to minimize.

**Transferability** A problem instance is defined by the graph $\mathcal{G}$, the number of servers $k$, and the arrival probabilities $\boldsymbol{p}$. Our goal is to learn a generalist policy $\pi^*(\mathcal{G}, k, \boldsymbol{p}; \boldsymbol{s}^t)$ that can perform well on a set of problem instances. Formally, we assume that there exists a probability distribution $\mathcal{D}$ from which each problem instance $\langle \mathcal{G}, k, \boldsymbol{p} \rangle$ is drawn. Our goal is to introduce a computational approach which given a set of training instances drawn from distribution $\mathcal{D}$, can find a generalist policy $\pi^*$ that performs as an optimal policy on any new test instance that is also drawn from $\mathcal{D}$.

## 3 PROPOSED COMPUTATIONAL APPROACH

### 3.1 BACKGROUND

**DQN with Experience Replay and Target Network** We employ the classical Deep Q-Network (DQN) approach as our learning algorithm (Mnih et al., 2013), incorporating two key techniques: experience replay and target network (Van Hasselt et al., 2016). Experience replay involves storing past experiences in a replay buffer and sampling mini-batches during the training process. This helps in breaking the temporal correlation in the sequence of observations and stabilizes the training. The target network is a separate network used to estimate target Q-values during training, which improves the stability of the learning process.

Our training process follows the standard DQN approach with target network and experience replay. Let $\mathcal{R}$ be the replay buffer containing experiences $\langle \boldsymbol{s}^t, a^t, r^t, \boldsymbol{s}^{t+1} \rangle$ sampled during interactions with the environment. The Q-network is trained iteratively to minimize the temporal difference error:

$$\mathcal{L}(\boldsymbol{\theta}) = \mathbb{E}_{(\boldsymbol{s}^t, a^t, r^t, \boldsymbol{s}^{t+1}) \sim \mathcal{R}} \left[ \left( Q\left(\boldsymbol{s}^t, a^t; \boldsymbol{\theta}\right) - r^t + \gamma \cdot \max_a Q\left(\boldsymbol{s}^{t+1}, a; \boldsymbol{\theta}^-\right) \right)^2 \right]$$

where $\gamma$ is a temporal discount factor, $\boldsymbol{\theta}$ and $\boldsymbol{\theta}^-$ are the parameters of the network and target network.

**Graph Convolutional Network**   To address the challenges posed by the large state and action spaces in our graph-based problem and to ensure transferability, we employ Graph Convolutional Networks (GCNs) (Kipf & Welling, 2017). GCNs allow us to exploit the relational information present in the graph structure, enabling more efficient and effective learning. We use graph convolutional layers to aggregate information from neighboring nodes, capturing the dependencies between nodes in the state representation.

## 3.2 Decomposition of Action-Value

A key challenge in the $k$-server problem is that action-values depend on the entire state, suggesting a need for aggregation over all nodes. However, such aggregation is difficult in large graphs. To address this challenge, we decompose the estimation of action-value $Q(\boldsymbol{s}, a)$ into two terms: *global value $Q^{global}(\boldsymbol{s})$*, which captures the overall value of the entire state, providing an estimate of the global context for the action-value estimation, and *local value $Q^{local}(\boldsymbol{s}_{x_a}^{local}, a)$*, which focuses on the specific server chosen by the action $a$ and its neighborhood $\boldsymbol{s}_{x_a}^{local}$, capturing local context that determines the advantage of choosing a particular action in a given state.

This decomposition allows for more efficient learning by separately processing global and local information. By combining classical DQN with GCN and a decomposition of action-values, we strive to develop a scalable and transferable solution for the stochastic $k$-server problem on graphs.

Note that for ease of presentation, we omit the superscript $t$ from the description of the neural-network architecture for our action-value estimator $Q(\boldsymbol{s}^t, a^t)$ as all calculations are performed for time step $t$.

## 3.3 Representation of Problem Instance and State

For the GCN, we represent the problem instance and state at time step $t$ as a matrix $\boldsymbol{S} \in \mathbb{R}^{n \times 3}$:

$$\boldsymbol{S} = \left[\mathbf{1}_{\{v \in \boldsymbol{x}\}}, \ \mathbf{1}_{\{v = \sigma\}}, \ \boldsymbol{p}\right], \tag{1}$$

where the first column $\mathbf{1}_{\{v \in \boldsymbol{x}\}}$ is an $n$-element indicator vector for the locations of the servers (i.e., 1 if there is a server on a node $v \in \mathcal{N}$, and 0 otherwise), the second column $\mathbf{1}_{\{v = \sigma\}}$ is an $n$-element indicator vector for the location of the request (i.e., 1 for the node $v$ where the request $\sigma$ is located, and 0 for all other nodes), and the third column is the vector $\boldsymbol{p}$ of request arrival probabilities. The above representation encompasses both the state, represented by $\mathbf{1}_{\{v \in \boldsymbol{x}\}}$ and $\mathbf{1}_{\{v = \sigma\}}$, and certain aspects of the problem instance, such as $\boldsymbol{p}$ (and implicitly the number of servers $k$). To complete the specification of the problem instance, we also provide adjacency matrix $\boldsymbol{A}$, which represents the topology $\mathcal{G}$, as input to our generalist policies.

To address scalability issues due to the magnitude of arrival probabilities varying with the size of the graph, we translate probabilities $\boldsymbol{p}$ to arrival rates $\Lambda = \{\lambda_1, \lambda_2, \ldots, \lambda_n\}$ by multiplying them with the number of nodes $n$: $\lambda_v = p_v \cdot n$. This transformation is necessary (as we verified empirically) to avoid the risk of perceiving arrival probabilities in large graphs as disproportionately low. By fixing the scale of the arrival probabilities via translating them to arrival rates, we ensure scalability.

## 3.4 Q-Network Architecture for Action-Value Decomposition

**Shared Backbone:** As shown in Figure 1, we use convolutional layers with residual connections (Kipf & Welling, 2017; He et al., 2016):

$$\boldsymbol{H}^{(l+1)} = \phi\left(\tilde{\boldsymbol{D}}^{-\frac{1}{2}} \tilde{\boldsymbol{A}} \tilde{\boldsymbol{D}}^{-\frac{1}{2}} \boldsymbol{H}^{(l)} \boldsymbol{W}^{(l)}\right) + \boldsymbol{H}^{(l)}, \tag{2}$$

where $\phi$ is a non-linear activation function, which we implement using ReLU in our experiments; $\tilde{\boldsymbol{A}}$ is the adjacency matrix of the graph $\mathcal{G}$ with added self-connections (i.e., $\tilde{\boldsymbol{A}} = \boldsymbol{A} + \boldsymbol{I}$, where $\boldsymbol{I}$ is the identity matrix); $\tilde{\boldsymbol{D}}$ is the diagonal matrix of node degrees (i.e., $\tilde{D}_{ii} = \sum_j \tilde{A}_{ij}$); and $\boldsymbol{W}^{(l)}$ is a matrix of trainable weights . The input to the first convolutional layer is the state representation $\boldsymbol{H}^{(0)} = \boldsymbol{S} \boldsymbol{W}^{init}$, where $\boldsymbol{W}^{init}$ represents an initial linear mapping applied to $\boldsymbol{S}$. This linear mapping

is introduced to enable the model to extract relevant features from the input before feeding it into the convolutional layers.

**Global Value:** There are $L$ such convolutions layers. The output $\boldsymbol{H}^{(L)}$ of the last convolutional layer is first used for estimating the global value $Q^{global}(\boldsymbol{s})$:

$$Q^{global}(\boldsymbol{s}) = \text{MLP}^{global}\left(\boldsymbol{H}^{(L)}\right), \tag{3}$$

where $\text{MLP}^{global}$ denotes a sequence of trainable fully-connected feed-forward layers with non-linear activation.

**Local Value:** Second, we use the output $\boldsymbol{H}^{(L)}$ of the last convolutional layer to estimate a local value $Q^{local}(\boldsymbol{s}^{local}_{x_a}, a)$ for each server $a \in \{1, \dots, k\}$:

$$Q^{local}(\boldsymbol{s}^{local}_{x_a}, a) = \text{MLP}^{local}\left(H^{(L)}_{x_a-}, d(\sigma, x_a)\right), \tag{4}$$

where $H^{(L)}_{x_a-}$ is row $x_a$ of $\boldsymbol{H}^{(L)}$, and $d(\sigma, x_a)$ is the distance between request $\sigma$ and node $x_a$. Note that $H^{(L)}_{x_a-}$ depends only on the $L$-hop neighborhood of $x_a$; hence, $\boldsymbol{s}^{local}_{x_a}$ is the subgraph spanned by nodes within $L$ hops of $x_a$.

**Action-Value Estimation** Finally, we calculate the action-value of sending server $a$ from node $x_a$ as

$$Q(\boldsymbol{s}, a) = Q^{global}(\boldsymbol{s}) + Q^{local}(\boldsymbol{s}^{local}_{x_a}, a). \tag{5}$$

Note that we can find the optimal action to take in state $\boldsymbol{s}$ based only on local values:

$$a^* = \max_a Q^{global}(\boldsymbol{s}) + Q^{local}(\boldsymbol{s}^{local}_{x_a}, a) = \max_a Q^{local}(\boldsymbol{s}^{local}_{x_a}, a). \tag{6}$$

This is crucial for scalability: once a policy is trained, the action-value estimator $Q^{local}(\boldsymbol{s}^{local}_{x_a}, a)$ can be evaluated on small subgraphs $\boldsymbol{s}^{local}_{x_a}$ to find an optimal action.

**Training Process:** To ensure transferability, we train the generalist policy over a set of randomly generated problem instances. This set includes instances drawn from a distribution $\mathcal{D}$, spanning diverse graph structures, server counts, and request arrival distributions. This approach ensures that the learned policy is transferable, i.e., it perform well on new instances from distribution $\mathcal{D}$.

## 4 NUMERICAL EVALUATION

### 4.1 BASELINE POLICIES

We compare our approach to several baseline approaches, including heuristics (greedy policy), baseline RL approach (MLP-based DQN), and state-of-the-art online algorithm (WFA). We provide a more detailed description of these baselines in Appendix C.

*Greedy Policy*: The Greedy policy simply selects the nearest server at each time $t$, choosing $a$ that minimizes the distance between the current request location $\sigma^t$ and the location of server $a$ at time $t$, i.e., $\pi_{\text{greedy}}(\boldsymbol{s}^t) = \arg\min_{a \in \mathcal{A}} d(\sigma^t, x^t_a)$. Despite its simplicity, it has historically demonstrated excellent performance (Bertsimas et al., 2019a).

*MLP DQN*: We also consider deep Q-learning (DQN) with a fully-connected feed-forward network (MLP) as a baseline, which we will refer to as MLP DQN. Note that MLP DQN is a stronger baseline than the approach proposed by Lins et al. (2019b). Please see Appendix C for a summary of the differences and for numerical results demonstrating that MLP DQN performs better than the approach proposed by Lins et al. (2019b). However, while MLP DQN is effective compared to prior approaches, it faces significant challenges in terms of scalability and transferability, which motivates our generalist GCN DQN approach.

*HARMONIC*: The Harmonic policy, a competitive algorithm for $k \geq 2$ in the $k$-server problem, achieves $O(2^k \log k)$-competitiveness against an adaptive online adversary (Bartal & Grove, 2000). The Harmonic policy prioritizes server selection based on inverse distance probabilities, i.e., $\pi_{\text{HARMONIC}}(\boldsymbol{s}^t) = \arg\max_{a \in \mathcal{A}} \frac{1}{d(\sigma^t, x^t_a)}$.

*BALANCE*: The BALANCE policy is chosen as a benchmark due to its k-competitive nature, making it suitable for evaluating algorithm performance (Bartal & Grove, 2000). It selects server $a$ at time $t$ to minimize cumulative distances from prior requests and the distance to the new request location, i.e., $\pi_{\text{BALANCE}}(\boldsymbol{s}^t) = \underset{a \in \mathcal{A}}{\arg\min} \left( \sum_{j=1}^{t-1} d(\sigma^j, x_a^j) + d(\sigma^t, x_a^t) \right)$.

*WFA*: Work Function Algorithm (WFA) is a server allocation optimization approach for a sequence of requests, modeled as a flow optimization problem Borodin & El-Yaniv (1998). WFA is one of the most well-established algorithmic approaches for solving the $k$-server problem, balancing theoretical competitiveness and practical performance (Rudec et al., 2009). WFA computes a decision based on historical data and the current request. To make a fair comparison with this approach, we provide a set of requests as *burn-in* for WFA, thereby enabling it to learn the request arrival distribution. We also conduct experiments without the burn-in for WFA; in these experiments, to make a fair comparison, the other approaches do not observe the actual arrival rates but rather arrival rates estimated from prior requests. Due to lack of space, we present these results in Appendix B.[2]

## 4.2 PROBLEM INSTANCES

**Graphs**  We evaluate our approach against benchmark algorithms on multiple classes of graphs.

*Random Grids*: We employ the GRE method introduced by Peng et al. (2012) and Peng et al. (2014) to generate two-dimensional grids that resemble real-world road networks. First, we create an $m \times m$ grid of nodes and edges ($n = m \cdot m$). For the sake of simplicity, we generate grids that are square shaped ($n = 9, 16, 25, 36, 49, 64, 81, 100$ nodes). Then, horizontal and vertical edges are randomly removed with different probabilities, and diagonal edges are randomly added. By using probabilities from Peng et al. (2014), we generate graphs that resemble real-world road networks in terms of their topology; we refer to the description from Peng et al. (2014) for details.

*Random Trees*: We generate random trees of a given size through an iterative process of adding nodes one-by-one. For each node $u$ (excluding the first), a parent node $v$ is chosen uniformly at random from the set of previously added nodes: $v \sim \text{Uniform}\{1, 2, \dots, u-1\}$. An edge is then added between the selected parent $v$ and the current node $u$, creating a random tree iteratively. For consistency, we generate random trees of the same sizes as the grids ($n = 9, 16, 25, 36, 49, 64, 81, 100$ nodes).

*Real-World Networks*: The Sioux Falls (SF) and the Eastern Massachusetts (EM) graphs represent real-world transportation networks with 24 and 74 nodes, respectively. The nodes represent intersections, while edges represent road segments between intersections (Transportation Networks for Research Core Team, 2023).

**Number of Servers and Arrival Rates**  For all problem instances, the number of servers is $k = \lfloor \frac{n}{6} \rfloor$. To generate request arrival probabilities $\boldsymbol{p}$, we draw random weights $w_1, w_2, \dots, w_n$ from an exponential distribution, and then normalize these weights: $p_i = \frac{w_i}{\sum_{j=1}^{n} w_j}$.

For each graph size, we generate 5 random grids and 5 random trees with different request arrival probabilities, resulting in 80 distinct problem instances. With the SF and EM graph topologies, we have 82 problem instances in total that we use for evaluation.

## 4.3 EXPERIMENTAL SETUP

**Model Training**  We individually train both our graph-specific policy (GCN DQN) and MLP DQN on each of the 82 problem instances. For MLP DQN, the training duration scales with the size of the graph. We initiate training for the smallest graph (size 9) with 300,000 steps, incrementing the number of steps by 20% for each subsequent graph size, which ensures convergence in our settings. Graph-specific policies, on the other hand, converge for all problem instances within 120,000 steps.

Furthermore, we train a generalist model (Generalist GCN DQN) designed to be agnostic to the specific number of nodes within a graph, applicable across all problem instances of that graph type. Specifically, for each graph size within a graph type (i.e., 9, 16, 25, 36, 49, 64, 81 and 100 nodes),

---

[2]Note we did not observe any significant advantage or disadvantage from using the *burn-in* approach.

we randomly generate 50 problem instances. After completing an episode of length 30 for all graph sizes, we generate a new set of 50 random problem instances, introducing both new topologies and new probabilities of arrival for nodes across all graph sizes. For SD and EM graph topologies, where the topology and graph size are fixed, we only modify node probability arrivals after each episode. Generalist GCN DQN models for tree and grid graphs converge within 960,000 steps, while those for EM and SF converge within 250,000 steps.

**Evaluation**  To evaluate the different policies, we generate 10 episodes for each problem instance, each containing 4000 requests. We use 100 requests for burn-in (see description of WFA above), and evaluate each approach on the remaining 3900 requests. We present performance with respect to an optimal offline algorithm. Grids and trees are evaluated based on sizes, resulting in 50 values for each graph type and size. For EM and SF networks with a single topology and graph size, we obtain 10 values each. To test the scalability of the Generalist GCN DQN, we conduct experiments on 10 episodes each for 5 grids and 5 trees of size 1024. Due to computational challenges, training GCN DQN, MLP DQN, and testing on the WFA (as explained later) on these large graphs is infeasible. Therefore, our evaluation on scalability uses the Greedy Policy as baseline.

**Hyper-Parameter Search**  WFA relies on a look-out window for past requests, with larger windows potentially leading to better performance. However, computational costs rise significantly for larger windows. Our affordable maximum window size for the WFA is set at 100, taking approximately 3 hours for a single episode and 30 hours for all episodes in a single EM problem instance. Meanwhile, training a generalist GCN DQN for grid graphs up to size 100 takes 16 hours. For the Generalist GCN DQN, key hyper-parameters include temporal discount (0.99), GCN layer parameters (12 layers, 128 hidden channels), and learning rate (0.001). Adjusting the temporal discount to 0.95 improves results for tree topology, while varying hidden channels, layer numbers and learning rate worsens results for all topologies. Also, the MLP applied to the last convolutional layer in the generalist model has hyper-parameters set at 7 layers and 32 hidden channels. Tuning these parameters impacts performance minimally. In the case of MLP DQN, we scale the number of parameters with graph size, maintaining parity with benchmarks and tuning temporal discount and learning rate.

**Hardware and Software**  We performed all experiments on a computer with an AMD EPYC 7763 64-core CPU and 1 TiB or RAM. We did not utilize a GPU for either training or evaluation due to the relatively small size of our neural networks. We implemented all neural networks using PyTorch. We implemented WFA following the most common approach, by casting it as a network flow problem Bertsimas et al. (2019b), and we used the NetworkX library to solve this problem, which is based on the network simplex algorithm (one of the widely used and highly efficient methods for solving the Minimum Cost Maximum Flow problem Kiraly & Kovacs (2012)).

**Inference Times**  To compare the inference times of the various approaches, we evaluated all of them on the real-world EM graph (72 nodes). For 10 episodes of 4,000 steps (40,000 steps in total), GCN DQN took 198.4 seconds, MLP DQN took 170.5 seconds, Greedy took 9.1 seconds, Balance took 18.6 seconds, and Harmonic took 69.0 seconds. Since all of these running times are below 0.01 seconds per decision, they can be considered negligible for most practical applications. However, WFA took approximately 30 hours in total, which is a notable disadvantage.

**Number of Parameters**  Note that the number of parameters in the MLP DQN architecture scales with the graph size, as the dimensionality of its input and output depends on the number of nodes $n$. For example, it has 13,608 parameters for $n = 9$ nodes and over 1.5 million for $n = 100$ nodes. On the other hand, the GCN architecture has a constant number of parameters, 228,866 in our experiments, regardless of the graph size.

## 4.4 NUMERICAL RESULTS

**Comparison of Policies**  As shown by Figure 2 and Table 1, MLP DQN initially performs well, but its performance significantly deteriorates with increasing graph size. Harmonic and Balance policies consistently underperform, while the Greedy policy shows competitiveness in specific instances but falls short of achieving the best performance. Notably, the Generalist GCN DQN not only outperforms all baseline policies, but it also surpasses a GCN DQN trained specifically for the given

Table 1: Travel Costs

| Type | Size $n$ | Harmonic | Balance | Greedy | WFA | MLP DQN | GCN DQN | Generalist GCN DQN |
|---|---|---|---|---|---|---|---|---|
| EM | 74 | 3.01 ± 0.04 | 1.79 ± 0.02 | 1.45 ± 0.02 | 1.39 ± 0.01 | 2.08 ± 0.03 | **1.32 ± 0.01** | 1.33 ± 0.02 |
| SF | 24 | 1.93 ± 0.02 | 1.48 ± 0.02 | 1.25 ± 0.00 | 1.25 ± 0.01 | 1.24 ± 0.01 | 1.22 ± 0.01 | **1.20 ± 0.01** |
| Grid (GRE) | 9 | 1.42 ± 0.03 | 1.32 ± 0.01 | 1.24 ± 0.05 | **1.21 ± 0.02** | 1.27 ± 0.07 | 1.25 ± 0.05 | 1.26 ± 0.05 |
| | 16 | 1.75 ± 0.04 | 1.47 ± 0.03 | 1.31 ± 0.03 | 1.29 ± 0.02 | 1.30 ± 0.03 | 1.25 ± 0.05 | **1.23 ± 0.03** |
| | 25 | 2.01 ± 0.03 | 1.55 ± 0.02 | 1.24 ± 0.02 | 1.24 ± 0.01 | 1.25 ± 0.03 | 1.19 ± 0.02 | **1.17 ± 0.01** |
| | 36 | 2.40 ± 0.08 | 1.63 ± 0.02 | 1.33 ± 0.03 | 1.29 ± 0.02 | 1.36 ± 0.07 | 1.24 ± 0.02 | **1.20 ± 0.02** |
| | 49 | 2.73 ± 0.06 | 1.69 ± 0.03 | 1.37 ± 0.03 | 1.33 ± 0.02 | 1.48 ± 0.07 | 1.32 ± 0.03 | **1.29 ± 0.01** |
| | 64 | 3.13 ± 0.05 | 1.71 ± 0.03 | 1.45 ± 0.03 | 1.35 ± 0.02 | 1.71 ± 0.11 | **1.29 ± 0.02** | 1.29 ± 0.02 |
| | 81 | 3.45 ± 0.07 | 1.73 ± 0.03 | 1.53 ± 0.04 | 1.39 ± 0.02 | 2.15 ± 0.12 | 1.36 ± 0.02 | **1.29 ± 0.01** |
| | 100 | 3.79 ± 0.06 | 1.75 ± 0.03 | 1.59 ± 0.04 | 1.37 ± 0.02 | 3.18 ± 0.36 | 1.32 ± 0.03 | **1.30 ± 0.02** |
| | 1024 | - | - | 1.45 ± 0.02 | - | - | - | **1.34 ± 0.02** |
| Random tree | 9 | 1.50 ± 0.06 | 1.38 ± 0.02 | 1.17 ± 0.09 | 1.21 ± 0.07 | 1.21 ± 0.14 | **1.15 ± 0.06** | 1.16 ± 0.09 |
| | 16 | 1.76 ± 0.07 | 1.53 ± 0.02 | 1.27 ± 0.05 | 1.31 ± 0.03 | 1.28 ± 0.03 | **1.22 ± 0.04** | 1.23 ± 0.04 |
| | 25 | 1.94 ± 0.06 | 1.59 ± 0.03 | 1.27 ± 0.03 | 1.33 ± 0.03 | 1.34 ± 0.04 | 1.27 ± 0.05 | **1.24 ± 0.02** |
| | 36 | 2.28 ± 0.08 | 1.71 ± 0.03 | 1.36 ± 0.04 | 1.39 ± 0.02 | 1.42 ± 0.04 | 1.30 ± 0.03 | **1.29 ± 0.03** |
| | 49 | 2.56 ± 0.14 | 1.81 ± 0.04 | 1.36 ± 0.04 | 1.41 ± 0.03 | 1.66 ± 0.11 | 1.35 ± 0.04 | **1.34 ± 0.03** |
| | 64 | 2.89 ± 0.17 | 1.87 ± 0.04 | 1.36 ± 0.04 | 1.42 ± 0.04 | 1.81 ± 0.12 | 1.34 ± 0.04 | **1.31 ± 0.02** |
| | 81 | 3.13 ± 0.17 | 1.92 ± 0.03 | 1.38 ± 0.03 | 1.45 ± 0.03 | 2.37 ± 0.20 | 1.36 ± 0.06 | **1.32 ± 0.03** |
| | 100 | 3.35 ± 0.17 | 1.96 ± 0.03 | 1.39 ± 0.03 | 1.46 ± 0.04 | 3.65 ± 0.27 | 1.43 ± 0.14 | **1.33 ± 0.02** |
| | 1024 | - | - | 1.41 ± 0.02 | - | - | - | **1.39 ± 0.02** |

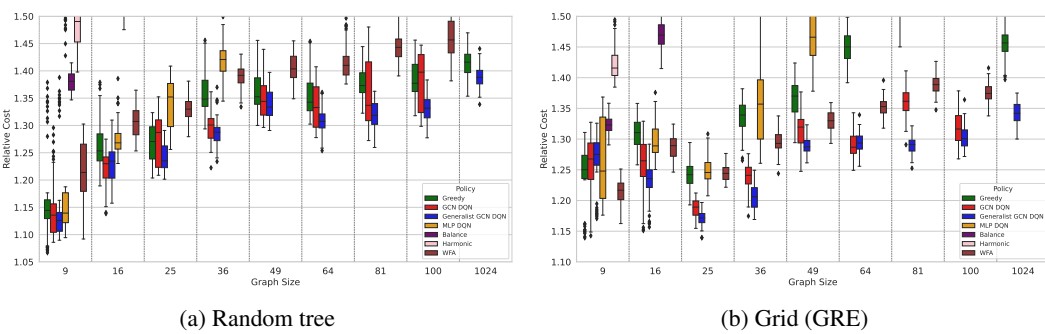

(a) Random tree          (b) Grid (GRE)

Figure 2: Episodic travel cost with various policies, expressed relative to the offline episodic cost (lower is better). The generalist GCN DQN approach outperforms all baselines comprehensively.

problem instance. The success of the Generalist GCN DQN may stem from its exposure to diverse graphs during training, enabling effective generalization.

**Zero-Shot Robustness to Distribution Shift in Arrival Rates** Figure 3 demonstrates that our proposed generalist policy is robust to significant changes in the distribution of arrival rates: a generalist policy that was trained on problem instances with arrival rates drawn from an exponential distribution performs well on problem instances with arrival rates drawn from lognormal, Poisson, and Bernoulli distributions, matching the performance of policies that were trained with these specific distributions.

**Ablation Study** We present an ablation study of our architecture in Appendix D, demonstrating the advantage of the global-local decomposition of the action-value function.

## 4.5 LIMITATIONS

Our focus is on the fundamental formulation of the online $k$-server problem, demonstrating strong performance and transferability for our approach. While we believe that our approach is applicable to specific practical problems (e.g., emergency-response dispatch), this will have to be confirmed by future experiments in practical environments based on complex simulations.

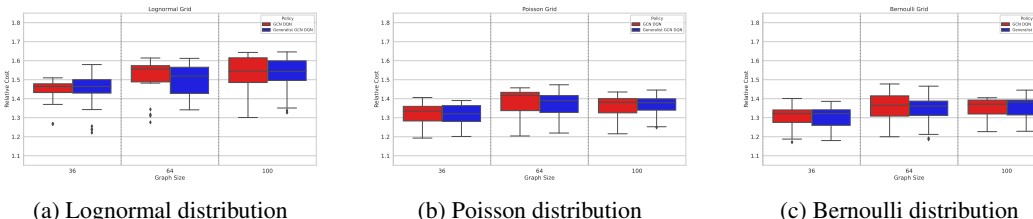

| (a) Lognormal distribution | (b) Poisson distribution | (c) Bernoulli distribution |

Figure 3: Generalist policy (blue) trained on problem instances with arrival probabilities drawn from an exponential distribution, evaluated on instances with arrival probabilities drawn from lognormal, Poisson, and Bernoulli distributions. The generalist policy is compared to specialist policies (red) trained with lognormal, Poisson, and Bernoulli distributions. All experiments are performed on GRE grid graphs; episodic travel costs are expressed relative to offline cost (lower is better). The results demonstrate that the generalist GCN DQN policy is robust to shifts in the distribution of arrival rates.

## 5 RELATED WORK

The $k$-server problem is one of the most widely studied online optimization problems in the last three decades, and providing a comprehensive summary of prior work is beyond the scope of this paper. We refer readers to prior work by Bertsimas et al. (2019b) and Koutsoupias (2009), who provide excellent overviews of related work in this domain. We present a relatively short account of prior work here (largely inspired by the detailed account given by Koutsoupias (2009)). The $k$-server problem was first introduced by Manasse et al. (1988; 1990). As Koutsoupias (2009) points out, the problem setting was proposed at a time when several important developments happened in the context of online algorithms. In particular, Sleator & Tarjan (1985) introduced the notion of competitive analysis (also known as "amortized efficiency"), a principled paradigm for evaluating online algorithms. This framework marked a rapid development in the field of online algorithms, including the introduction of the $k$-server problem (Manasse et al., 1988). The problem has remained a popular choice among computer scientists over the last three decades as it presents a rather simple setting that serves as an abstraction for many practical problems (e.g., emergency resource allocation and caching (Bertsimas et al., 2019b) while presenting several algorithmic challenges. Much of prior work has focused on designing online algorithms, evaluating their competitive ratios, and generalizing the competitive ratios of online algorithms (Chrobak et al., 1991; Chrobak & Larmore, 1991; Grove, 1991). We present a more detailed description of offline and online versions of the $k$-server problem and prior work (albeit limited) on deep reinforcement learning to solve the $k$-server problem in Appendix A.

## 6 CONCLUSION

Despite the fundamental nature and significance of the $k$-server problem, there is extremely limited prior work on solving this problem using reinforcement learning, even though learning has obvious advantages (e.g., evaluation of WFA on a few instances took significantly longer than training our DRL policies). We hypothesize that this gap is due to the difficulty of handling the large and combinatorial state and action spaces of the $k$-server environment. We addressed this gap by introducing a novel approach that decomposes action-values into a global value and local advantage, estimated from a shared graph convolutional network. Not only did this approach scale to larger graphs than any prior learning-based approach, but we also demonstrated that the resulting generalist policies can be transferred to other instances with no further training. In fact, generalist policies outperformed graph specific ones—an interesting phenomenon, which we explain with generalist policies being forced to generalize more effectively during training.

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

Table 2: List of Symbols

| Symbol | Description |
|---|---|
| | *k*-Server Problem |
| $\mathcal{G} = (\mathcal{N}, \mathcal{E})$ | graph with set of nodes $\mathcal{N}$ and set of edges $\mathcal{E}$ |
| $d(u, v)$ | shortest path distance between nodes $u$ and $v$ |
| $n$ | number of nodes ($n = |\mathcal{N}|$) |
| $k$ | number of servers ($k \in \mathbb{N}$) |
| $p_v$ | probability of a request arising at node $v \in \mathcal{N}$ |
| $T$ | time horizon ($T \in \mathbb{N}$) |
| $\sigma^t$ | location of the request at time $t$ ($\sigma^t \in \mathcal{N}$) |
| $x_i^t$ | location of server $i$ at time $t$ ($x_i^t \in \mathcal{N}$) |
| | Markov Decision Process |
| $\mathcal{S}$ | set of states ($\mathcal{S} \ni s^t = (\boldsymbol{x}^t, \sigma^t)$) |
| $\mathcal{A}$ | set of actions ($\mathcal{A} = \{1, \dots, k\}$) |
| $\rho$ | state transition function ($\rho : \mathcal{S} \times \mathcal{A} \times \mathcal{S} \mapsto \mathbb{R}$) |
| $r$ | reward function ($r(s_t, a) = d(\sigma^t, x_a^t)$) |
| $\pi$ | policy function ($\pi : \mathcal{S} \mapsto \mathcal{A}$) |

## A  EXTENDED DISCUSSION OF RELATED WORK

*k*-**server Problem**  There are two major types of the $k$-server problem—the offline version, in which the set of requests to be served is known beforehand, and the online version, in which the requests arrive as decisions are made. The offline problem can be reduced to the standard network flow optimization problem and solved efficiently (Chrobak et al., 1991; Bertsimas et al., 2019b). The reduction centers around adding a source and a sink to to the existing graph, along with an additional node for each of the requests, manipulating the weights of the edges to direct the flow between the source and the sink, and solving a minimal-cost maximal-flow problem problem on the modified graph (Chrobak et al., 1991). The offline solution can then be used for competitive analysis of the online problem, where the requests are not known in advance. Many online approaches have been proposed for solving the $k$-server problem, and despite the complexity of the problem, several simple algorithms work reasonably well. For example, a simple greedy algorithm is a deterministic approach that always dispatches the server closest to a request for service. Although non-competitive, such an approach has been noted to work well in several settings (Bertsimas et al., 2019b). As opposed to deterministic approaches, there are several randomized algorithms that work well; e.g., HARMONIC is a competitive algorithm that selects servers with probability inversely proportional to the distance from the request (Raghavan & Snir, 1989). In an orthogonal categorization of the problem type, Dehghani et al. (2017) introduced the stochastic $k$-server problem, where the requests are drawn from a (potentially time-varying) probability distribution; this setting mimics many real-world problems such as emergency response and micro-transit optimization, where a distribution over the arrival of requests can be approximated using historical data.

**Deep Reinforcement Learning**  The sequential nature of the $k$-server problem naturally leads to it being modeled as a stochastic control process. Surprisingly, this modeling paradigm has been under-explored in prior work. The earliest effort was made by Junior et al. (2005), who used reinforcement learning (Q-learning) to compute the optimal decisions for small problem instances. Lins et al. (2019a) transformed the $k$-server problem to a visual task problem and used reinforcement learning, but neither approach can scale to large problem instances. Our work presents the first decision-theoretic approach to model the $k$-server problem that can scale to large problem instances. Moreover, we exploit the information embedded in the local structure of graphs to learn generalist policies that perform well even when the graph topology or the arrival distribution of the requests changes (as long as the change is captured during training of the policy). We point out that such generalist policies are particularly desirable in practice. Consider emergency response as an example, where the underlying graph denotes the road network of a city. Roads might get closed or be affected by traffic congestion, thereby changing the underlying network Pettet et al. (2021). While

Table 3: Travel Costs with Unknown Arrival Probabilities

| Type | Size $n$ | Greedy | WFA | GCN DQN | Generalist GCN DQN |
|---|---|---|---|---|---|
| EM | 74 | 1.44 ± 0.02 | 1.39 ± 0.01 | 1.35 ± 0.02 | **1.34 ± 0.02** |
| SF | 24 | 1.28 ± 0.01 | 1.28 ± 0.01 | **1.22 ± 0.01** | **1.22 ± 0.01** |
| Grid (GRE) | 9 | 1.24 ± 0.05 | **1.21 ± 0.02** | 1.26 ± 0.05 | 1.26 ± 0.05 |
| | 16 | 1.31 ± 0.03 | 1.29 ± 0.02 | 1.25 ± 0.05 | **1.23 ± 0.03** |
| | 25 | 1.24 ± 0.02 | 1.25 ± 0.01 | 1.19 ± 0.02 | **1.17 ± 0.01** |
| | 36 | 1.33 ± 0.03 | 1.29 ± 0.02 | 1.24 ± 0.02 | **1.21 ± 0.02** |
| | 49 | 1.37 ± 0.03 | 1.33 ± 0.02 | 1.32 ± 0.03 | **1.29 ± 0.01** |
| | 64 | 1.45 ± 0.03 | 1.35 ± 0.01 | **1.29 ± 0.02** | 1.30 ± 0.02 |
| | 81 | 1.53 ± 0.04 | 1.39 ± 0.01 | 1.37 ± 0.03 | **1.29 ± 0.01** |
| | 100 | 1.59 ± 0.04 | 1.38 ± 0.02 | 1.32 ± 0.02 | **1.31 ± 0.02** |
| Random tree | 9 | 1.17 ± 0.09 | 1.21 ± 0.07 | **1.15 ± 0.06** | 1.16 ± 0.09 |
| | 16 | 1.27 ± 0.05 | 1.31 ± 0.03 | **1.23 ± 0.04** | **1.23 ± 0.04** |
| | 25 | 1.27 ± 0.03 | 1.33 ± 0.03 | 1.28 ± 0.05 | **1.24 ± 0.03** |
| | 36 | 1.36 ± 0.04 | 1.39 ± 0.02 | 1.31 ± 0.03 | **1.29 ± 0.03** |
| | 49 | 1.36 ± 0.04 | 1.41 ± 0.03 | 1.36 ± 0.04 | **1.34 ± 0.03** |
| | 64 | 1.36 ± 0.04 | 1.42 ± 0.04 | 1.34 ± 0.04 | **1.31 ± 0.02** |
| | 81 | 1.38 ± 0.03 | 1.45 ± 0.03 | 1.36 ± 0.06 | **1.33 ± 0.02** |
| | 100 | 1.39 ± 0.03 | 1.46 ± 0.04 | 1.44 ± 0.14 | **1.34 ± 0.03** |

existing approaches will require time for retraining (a luxury not available in time-critical domains such as emergency response), a generalist approach can easily adapt to the changed conditions.

# B NUMERICAL EVALUATION WITH UNKNOWN ARRIVAL PROBABILITIES

Recognizing that the WFA requires substantial exposure to requests before demonstrating optimal performance Bertsimas et al. (2019a), we implement a fair comparison strategy by introducing a burn-in period of 100 requests for the WFA, all arriving under the same distribution, and present these results in the main text. However, acknowledging potential reader skepticism about providing an unfair advantage to other methods, as they possess information about arrival probabilities from the first request, we address this concern by conducting additional experiments using the Estimated Arrival Probabilities Approach.

*Estimated Arrival Probabilities Approach*: This approach involves comparing other methods with the WFA while incorporating estimated arrival probabilities. Initially, precise information about the arrival probabilities of requests originating from specific nodes is unavailable. The arrival probabilities for the first request from all nodes are assumed to be zero. As requests are processed, these probabilities are iteratively updated. This iterative process continues as more data is collected.

The estimated arrival probability at time $t$ for a request originating from node $j$ can be denoted as $\hat{p}_j^t$. The iterative update process for the estimated arrival probabilities can be expressed as follows:

$$\hat{p}_j^t = \frac{\sum_{i=1}^{t} I(\sigma^i = j)}{t}$$

where $I(\sigma^i = j)$ is an indicator function that equals 1 if the $i$-th request originated from node $j$, and 0 otherwise. This formula updates the estimated arrival probability based on the history of observed requests.

This iterative update continues as more requests are processed over time, providing a dynamic estimation of arrival probabilities and eventually converging to the real probability distribution of a particular problem instance. Then, using the same method described in the methodology part, we transform these probabilities to arrival rates.

## B.1 NUMERICAL RESULTS

In this section, we compare the WFA against GCN DQN and Generalist GCN DQN, as the latter approaches consider request arrival probabilities during estimation. Additionally, we include the

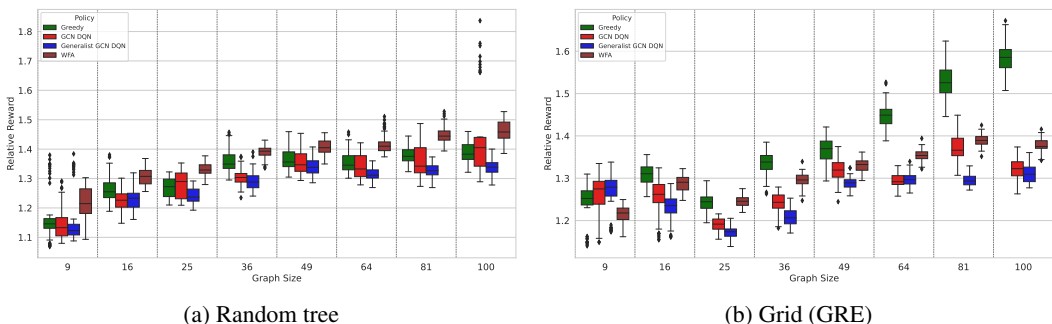

(a) Random tree           (b) Grid (GRE)

Figure 4: Episodic travel cost of various policies with unknown arrival probabilities.

Table 4: Comparison of MLP DQN and Lins et al.'s Approach

| Type | Size $|\mathcal{V}|$ | MLP DQN | Lins et al. (2019b) |
|------|------|---------|---------------------|
| Grid | 16 | **1.30 ± 0.03** | 1.58 ± 0.17 |
|      | 49 | **1.48 ± 0.07** | 2.81 ± 0.37 |
|      | 100 | **3.18 ± 0.36** | 6.74 ± 0.79 |
| Tree | 16 | **1.28 ± 0.03** | 1.37 ± 0.17 |
|      | 49 | **1.66 ± 0.11** | 2.23 ± 0.35 |
|      | 100 | 3.65 ± 0.27 | **3.64 ± 0.39** |

Greedy policy. The results, shown in Figure 4, once again highlight the superior performance of Generalist GCN DQN, outperforming GCN DQN.

## C    EXTENDED DESCRIPTION OF BASELINE POLICIES

We compare our proposed approach against several baselines, which represent a wide spectrum of strategies and algorithms for addressing the $k$-server problem, ranging from simple heuristics to RL-based approaches.

*Greedy Policy*: The greedy policy is a widely used benchmark in the $k$-server problem the idea of which is to send the nearest available server to the request Bertsimas et al. (2019b). To elaborate, at time $t$, it selects the server action $a$ that minimizes the shortest path distance between the current request location $\sigma^t$ and the location of server $a$ at time $t$, $x_a^t$:

$$\pi_{\text{greedy}}(\boldsymbol{s}^t) = \arg\min_{a \in \mathcal{A}} d(\sigma^t, x_a^t) \tag{7}$$

This policy aims to minimize the travel distance between the servers and the current request, making a locally optimal decision at each time step.

*MLP DQN*: Our MLP DQN baseline is similar to the approach proposed by Lins et al. (2019b) since both are trained using deep Q-learning; however, our MLP DQN baseline is built on a more capable architecture. Lins et al. (2019b) propose an MLP with only 1 hidden layer having 64 to 1024 neurons with sigmoid activation. We performed a hyperparameter search to find a better architecture; in our experiments, we use an MLP with 2 hidden layers having $12 \times n$ neurons with ReLU activation. Further, we represent the state using two $n$-element vectors, $\mathbf{1}_{\{v \in \boldsymbol{x}\}}$ and $\mathbf{1}_{\{v = \sigma\}}$ (i.e., 1-hot encoding the server and request locations), while Lins et al. (2019b) propose to represent the state using a single $n$-element vector: $\mathbf{1}_{\{v \in \boldsymbol{x}\}} - 0.5 \times \mathbf{1}_{\{v = \sigma\}}$. Table 4 compares the MLP DQN policy to the approach proposed by Lins et al. (2019b) in term of the total episodic travel time (lower is better) on grid and tree graphs of various sizes. Both were trained and evaluated on the same sets of graphs. The results demonstrate the MLP DQN either significnatly outperforms or matches the performnance of Lins et al. (2019b).

*HARMONIC*: The Harmonic is an algorithm that for $k \geq 2$ for the $k$-server problem is $O(2^k \log k)$-competitive against an adaptive online adversary Bartal & Grove (2000). It works by deciding which

server should handle a new request based on their distances from the request location. Servers closer to the request location have a higher chance of being chosen. The algorithm calculates the probability for each server to be selected, and this probability is influenced by the inverse of their distance from the request location. The normalization factor helps ensure that these probabilities are correctly balanced.

The Harmonic policy, denoted by $\pi_{\text{HARMONIC}}$, can be expressed as:

$$\pi_{\text{HARMONIC}}(\boldsymbol{s}^t) = \arg\max_{a \in \mathcal{A}} \frac{1}{d(\sigma^t, x_a^t)}$$

In words, the Harmonic policy at time $t$ selects the server action $a$ that maximizes the inverse of the distance from the current request location $\sigma^t$ to the location of server $a$ at time $t$, $x_a^t$. This probability-based approach aims to favor servers that are closer to the request location, with the normalization factor ensuring correct balance of probabilities.

*BALANCE*: BALANCE is a deterministic algorithm aimed at maintaining roughly equal total distances traveled by all servers. The interest in using the BALANCE algorithm as a benchmark stems from its k-competitive nature, making it a reasonable choice for evaluating the performance of algorithms and its simplicity making it a suitable baseline for comparison against more complex algorithms Bartal & Grove (2000).

The algorithm works by selecting the server action $a$ that minimizes the combined distance traveled from previous requests and the distance to the new request location. Mathematically, this can be expressed as:

$$\pi_{\text{BALANCE}}(\boldsymbol{s}^t) = \arg\min_{a \in \mathcal{A}} \left( \sum_{j=1}^{t-1} d(\sigma^j, x_a^j) + d(\sigma^t, x_a^t) \right)$$

In words, the BALANCE policy at time $t$ selects the server action $a$ that minimizes the sum of distances traveled by server $a$ for all previous requests up to time $t - 1$ and the distance from the current request location $\sigma^t$ to the location of server $a$ at time $t$, $x_a^t$. This algorithm aims to distribute the workload among servers, maintaining roughly equal total distances traveled.

*WFA*: The Work Function Algorithm (WFA) Borodin & El-Yaniv (1998) is a server allocation optimization approach for a sequence of requests, modeled as a flow optimization problem. We use the WFA because it stands out as one of the most important online algorithms for the K-server problem, addressing both theoretical competitiveness and practical performance concerns Rudec et al. (2009). Distinguished by its utilization of comprehensive historical data, the WFA contrasts with simpler online methods in making informed online decisions Bertsimas et al. (2019a).

The algorithm constructs a directed graph, where nodes represent sources (e.g., initial server locations, requests) and sinks (e.g., the final server configuration, the end of the process). Edges in the graph are assigned capacities and weights to depict the flow of resources and associated costs. The algorithm seeks to find the optimal flow through the graph, minimizing the overall cost by balancing offline and online optimization terms.

Since there is a bottleneck with the WFA in the sense that it needs to be subjected to a substantial number of requests before it exhibits optimal performance, we have to establish a fair comparison with it. Therefore, the comparison between the WFA and alternative methods commences after the WFA has undergone exposure to a predefined number of requests (100 requests) that arrive according to the same distribution. The established burn-in period ensures that the comparison accurately reflects the WFA's capabilities, allowing it adequate exposure for effective evaluation. This adaptation is crucial for the WFA because real-time decisions are made based on the current state of the system and incoming requests. While acknowledging that some primary approaches may have an advantage by knowing arrival probabilities from the first request, we further explore this scenario in Appendix B, comparing other methods with the WFA while incorporating estimated arrival rates.

## D ABLATION STUDY

There are three major components of our approach: the graph convolutional layers, the global estimator $Q^{\text{global}}$, and the local estimator $Q^{\text{local}}$. The convolutional layer cannot be omitted since the

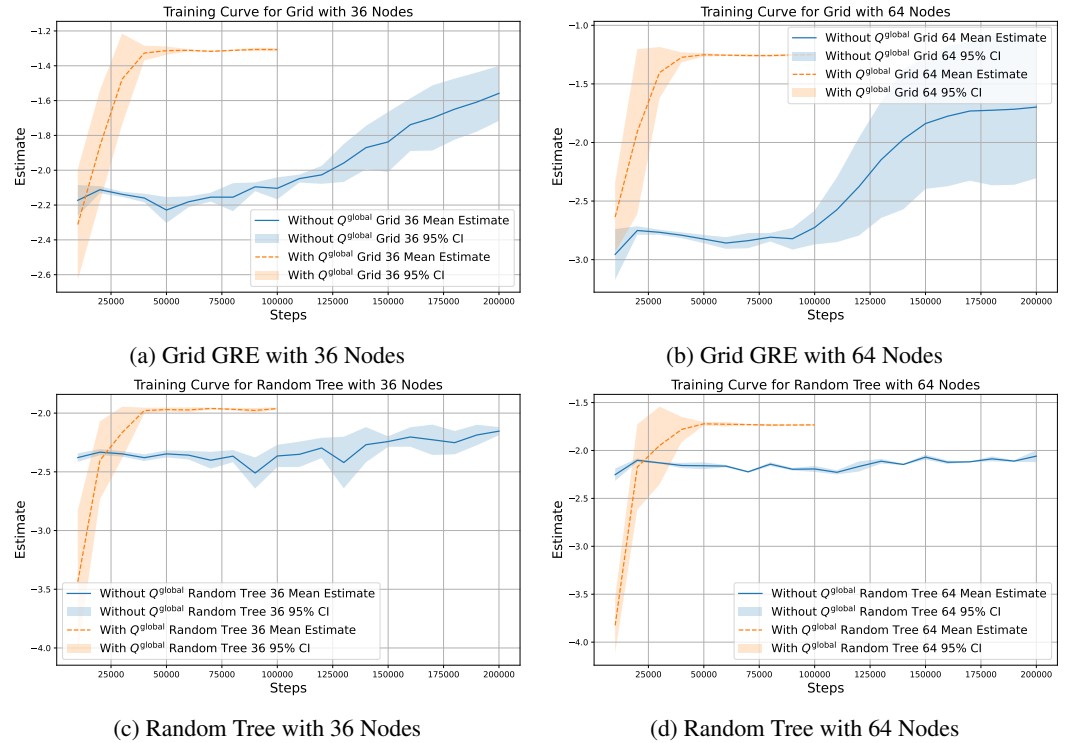

(a) Grid GRE with 36 Nodes        (b) Grid GRE with 64 Nodes

(c) Random Tree with 36 Nodes        (d) Random Tree with 64 Nodes

Figure 5: Learning curves for our proposed approach (with $Q^{global}$) and its ablation (without $Q^{global}$). Each figure shows learning curves for a particular problem instance (i.e., particular topology and arrival rates); vertical axis is the travel cost multiple by minus one (higher values are better). For each problem instance, we trained both the proposed approach and its ablation multiple times with random weight initialization: solid lines represent the averages of these runs, while shaded areas represent their 95% confidence intervals. The results demonstrate the importance of $Q^{global}$: without it, learning is significantly slower and often converges to suboptimal policies.

global-local decomposition would then be impossible, and there would be nothing left of our proposed approach. The local estimator $Q^{\text{local}}$ cannot be omitted since this would make the action-value estimates $Q(\boldsymbol{s}, a)$ independent of the action $a$. However, it is possible to omit the global estimator $Q^{\text{global}}$. To demonstrate the importance of including the global estimator $Q^{\text{global}}$ in our approach, we conducted an ablation study. Figure 5 shows learning curves for out proposed approach and its ablation (higher values are better). Our results clearly show that the global estimator leads to faster convergence and better policies.

