# OpenReview forum: "Generalist Policy for k-Server Problem on Graphs using Deep Reinforcement Learning with Action-Value Decomposition"
_ICLR.cc/2025/Conference — ICLR 2025 Conference Withdrawn Submission_

### Official Review · Reviewer_QBut · 2024-11-02

**Soundness:** 2
**Presentation:** 2
**Contribution:** 1
**Rating:** 3
**Confidence:** 4

**Summary:**

This paper addresses the k-server problem, a fundamental optimization challenge where k mobile servers must be efficiently dispatched to serve requests arriving at nodes in a graph. The authors introduce a  deep reinforcement learning approach that decomposes the action-value function into global and local components, both estimated using a shared graph convolutional network backbone. This architecture enables learning of "generalist" policies that can transfer across different graph topologies and request distributions without retraining. The approach significantly outperforms traditional baselines including the Work Function Algorithm and greedy policies, while showing better scalability than previous deep learning methods.

**Strengths:**

1. The problem investigated in this paper is interesting and important.

2. The objective of training a generalist policy that can solve any k-server problems is also interesting.

**Weaknesses:**

1. The technical contribution of this paper is weak. With GNN as the problem representation and RL as the optimization method is not a novel idea, examples include (https://arxiv.org/abs/2106.04927) and (https://arxiv.org/abs/2405.03518). So what is the technical contributions in terms of the representations of the problem and the training methods?

2. Besides, DQN, i.e., Q-learning with target network, is also not a new method, some advanced methods such as Distributional DQN and RAINBOW are proposed for better results. as well as the actor-critic methods, e.g., SAC and PPO.  So why not try more advanced methods for this problem?

3. While the authors show some transfer between different arrival distributions, they don't thoroughly analyze what properties of graphs/distributions enable successful transfer. The generalization capabilities seem primarily empirical without theoretical backing.

4. The ablation study only examines removing the global Q-value component. There's no investigation of: the training distributions, the training methods, e.g., RL methods, the decomposition approaches.

5. Theoretical results are missing. If ignoring the theoretical analysis, the k-server is not a difficult problem, compared with other complex problems solved by DRL methods.

**Questions:**

You can find more details about the questions in the weakness section.

The questions I want the authors to address include:
1. Could you provide some theoretical results about the proposed methods?

2. Ablations on the RL methods, decomposition approaches, the training distribution over problem instances.

3. Could you summarize the technical contributions of the method?

---

### Official Review · Reviewer_WxwF · 2024-11-02

**Soundness:** 2
**Presentation:** 2
**Contribution:** 2
**Rating:** 3
**Confidence:** 4

**Summary:**

The paper addresses the stochastic k-server problem using a reinforcement learning approach where they use the idea of GCN and pre-training on a distribution of graphs to achieve a generalist policy. The learned policy can scale to graphs with ~1000 nodes and transfer to unseen graphs (from the distribution as the training graphs).

While the adoption of RL and GCN for the stochastic k-server problem has some novelty, the idea in this paper is almost identical to a rich line of works that use RL plus graph embedding like GCN for combinatorial optimization problems (e.g., surveys [1] [2]), which unfortunately has not been discussed.

[1] Mazyavkina, Nina, et al. "Reinforcement learning for combinatorial optimization: A survey." Computers & Operations Research 134 (2021): 105400.

[2] Bengio, Yoshua, Andrea Lodi, and Antoine Prouvost. "Machine learning for combinatorial optimization: a methodological tour d’horizon." European Journal of Operational Research 290.2 (2021): 405-421.

---

**Post-rebuttal**: there does not seem to be a rebuttal from the authors, so I'll keep my score.

**Strengths:**

- The stochastic k-server problem is well-motivated
- The idea of applying GCN and RL for the stochastic k-server problem is somewhat novel

**Weaknesses:**

- While the adoption of RL and GCN for the stochastic k-server problem has some novelty, the idea in this paper is almost identical to a rich line of works that use RL plus graph embedding like GCN for combinatorial optimization problems (e.g., surveys [1] [2]), which unfortunately has not been discussed. In most of these works, e.g., [3],[4] where the idea is almost identical to the so-called "generalist policy" in this paper, the RL policy is represented using a type of graph embedding (e.g., Struct2vec in [3], GAT in [4]), and RL policies are trained over a distribution of graphs, and tested on new graphs from the same distribution.
- The performance of the "generalist GCN-RL" is very close to that of GCN-RL.

[1] Mazyavkina, Nina, et al. "Reinforcement learning for combinatorial optimization: A survey." Computers & Operations Research 134 (2021): 105400.

[2] Bengio, Yoshua, Andrea Lodi, and Antoine Prouvost. "Machine learning for combinatorial optimization: a methodological tour d’horizon." European Journal of Operational Research 290.2 (2021): 405-421.

[3] Khalil, Elias, et al. "Learning combinatorial optimization algorithms over graphs." Advances in neural information processing systems 30 (2017).

[4] Kool, Wouter, Herke van Hoof, and Max Welling. "Attention, Learn to Solve Routing Problems!." International Conference on Learning Representations. 2018.

**Questions:**

NA

---

### Official Review · Reviewer_EiVM · 2024-11-02

**Soundness:** 2
**Presentation:** 3
**Contribution:** 3
**Rating:** 5
**Confidence:** 3

**Summary:**

The paper explores a scalable reinforcement learning approach for solving the k-server problem on graphs. The authors introduce a deep reinforcement learning (RL) model that generalizes across different graph structures without retraining, making it adaptable and scalable. The proposed solution decomposes action-value calculations into a global term (capturing the broader state) and a local term (focused on the chosen server's vicinity), which is key to scalability. The model leverages a graph convolutional network (GCN) architecture, enabling it to handle the complexity of graph-based k-server problems. The results show that the generalist GCN-based RL model outperforms various baselines, including the Work Function Algorithm (WFA) and simpler heuristic methods, in terms of travel cost and adaptability.

**Strengths:**

The RL method proposed by the authors is quite scalable and can solve large-scale K-solver problems with high accuracy.

**Weaknesses:**

The interpretability of the results is limited. Specifically, there is no policy analysis of the generalist policy's superiority and why it beats all the well-known baseline algorithms (especially WFA). Also, the authors report total travel costs as a result with no estimate of the optimality gaps. To further understand the seeming superiority of the RL approach, I would also suggest the authors report the results for offline K-servers when the arrival information is fully specified (and the arrival distribution can change over time). These are reasons to make the evaluation of the soundness of the computational results hard.

**Questions:**

As mentioned, I view the major weakness of the paper as a lack of interpretability. To address this issue, I suggest the authors provide a more detailed policy analysis, convert their computational results into optimality gaps (or competitive ratios), and provide results for offline K-servers.

---

### Official Review · Reviewer_gy2d · 2024-11-04

**Soundness:** 1
**Presentation:** 1
**Contribution:** 1
**Rating:** 3
**Confidence:** 4

**Summary:**

In this work, the authors propose a scalable and transferable reinforcement learning-based algorithm for solving the stochastic k-server problem in an online setting. Their method combines two key approaches: (1) using a graph convolutional network as the primary neural network architecture within a deep Q-network (DQN) and (2) introducing a novel decomposition of the action-value function to ensure fast convergence. The authors also present experiments to further support their findings. These experiments evaluate their method, termed "Generalist GCN DQN," across various graph and tree structures and compare it against other algorithms from different perspectives for the well-known k-server problem.

**Strengths:**

1. This work presents an interesting effort by the authors to model and solve a variant of the k-server problem using a reinforcement learning framework.

2. The authors offer valuable insights by comparing a learning-based approach with non-learning methods for the k-server problem.

**Weaknesses:**

I will first address my concerns with the technical aspects, followed by comments on writing and presentation. I hope the authors find these suggestions and insights helpful.

Technical Perspective:
Section 3.2 (Decomposition of Action-Value): The purpose of the decomposition approach is unclear. Specifically, the terms $Q^{global}(s)$ and $Q^{local}(s^{local}_{x_a}, a)$ are not clearly explained. I encourage the authors to provide a more formal and mathematical definition of these global and local contexts rather than relying on natural language. Additionally, equations (3) and (4) require further clarification. In equation (5), it is unclear why this approximation is valid and how much the estimated $Q(s,a)$ differs from the true value estimation under a given policy $\pi$.

Graph Convolutional Network (GCN) with Deep RL Methods: The application of GCNs with Deep RL methods is not novel, and it seems the authors overlooked relevant prior work, specifically "Jiang, Jiechuan, et al., 'Graph Convolutional Reinforcement Learning,' arXiv: Learning (2018)." It would be helpful to understand how the proposed architecture compares with the Deep Graph Network (DGN) algorithm from that work.

Experiments and Evaluation: Three key baselines—Harmonic, MLP DQN, and Balance—are missing from Table 3 (for unknown arrival probabilities). Additionally, there is a rich literature on DQN improvements, but other enhanced versions of DQN (such as PER or Rainbow) are not included in comparisons. It is also worth noting that Vanilla DQN shows comparable performance to "Generalist GCN DQN" on smaller graphs and trees, as seen in Table 1. Moreover, it is unclear which methods require the "burn-in" process for estimating arrival probabilities, as there are contradictory statements: in the WFA section, it is mentioned that "burn-in" is needed, yet other sections indicate its use across all methods. It should also be clarified for which methods the arrival probability is known beforehand and for which it is not. Furthermore, if arrival distributions are known, as suggested in the Introduction, why is additional calculation required?

Objective in Problem Setting (Section 2.1): The objective of the problem is missing in Section 2.1. While the authors mention it informally in the Introduction, the objective is not formally defined within the problem setting.

Evaluation Metrics: While the authors focus on "low computational effort" (i.e., cost as indicated in the tables), an important aspect is missing: how the training process differs among the three learning baselines. While computational efficiency post-training is crucial, the computational effort during training is equally significant. There is a trade-off here, as MLP DQN likely requires substantially less computation during training compared to the authors’ advanced model.

Ablation Study: The authors do not assess the individual impact of the decomposition and GCN components, claiming it is impossible. If this is indeed the case, it implies that the decomposition method is only applicable within the GCN architecture, reducing its general applicability. Given the architecture in Figure 1, one could implement the decomposition idea by freezing layers after $H^{(L)}$ and modifying only the previous layers, thereby testing the decomposition approach with other architectures.

Missing Baselines: Important baselines, such as the method from "Khadiev, Kamil R., and Maxim Yagafarov, 'The Fast Algorithm for Online k-server Problem on Trees,' Computer Science Symposium in Russia (2020)," are missing. A comparison with this algorithm would provide valuable context.

Experimental Analysis: The discussion of experimental observations lacks sufficient detail. For instance, in Table 3, what would occur if $n > 100$? As $n$ approaches 100, the GCN DQN value converges toward WFA; however, it is unclear how performance scales beyond this. Additionally, there is no discussion of instance difficulty. Figure 3, illustrating robustness, does not include other values of $n$, and the reasoning for this omission is unclear.

Theoretical Analysis: The paper lacks theoretical analysis, particularly concerning the decomposition approach.

Writing and Presentation Perspective:
Problem Setting Details (Section 2.1): Key details, such as the problem objective and assumptions (e.g., online vs. offline setting), are not well-presented and are scattered across different sections.

Notation Consistency: Some notations related to reinforcement learning are inconsistent; for example, $\pi^*$ is initially presented as a function of a single state but later appears as a function of four inputs.

Clarity in Terminology: When discussing efficiency, specify whether it refers to computational efficiency or sample efficiency. Several terms in the paper are vague. For example, in the transferability discussion, the authors state "perform as an optimal policy"; if this is indeed the case, the suboptimality error should be provided.

Important reference to main DQN paper is missing

**Questions:**

I already mentioned my question in the Weakness related comments.

---

### Note · Authors · 2024-12-03

I have read and agree with the venue's withdrawal policy on behalf of myself and my co-authors.